

# Indistinguishability under adaptive chosen-ciphertext attack secure double-NTRU-based key encapsulation mechanism

Kübra Seyhan[1,2] and Sedat Akleylek[1,2,3]

[1] Department of Computer Engineering, Ondokuz Mayis University, Samsun, Turkey
[2] Cyber Security and Information Technologies Research and Development Centre, Ondokuz Mayis University Samsun, Samsun, Turkey
[3] Chair of Security and Theoretical Computer Science, University of Tartu, Tartu, Estonia

## ABSTRACT

In this article, we propose a double-NTRU (D-NTRU)-based key encapsulation mechanism (KEM) for the key agreement requirement of the post-quantum world. The proposed KEM is obtained by combining one-way D-NTRU encryption and Dent's KEM design method. The main contribution of this article is to construct a D-NTRU-based KEM that provides indistinguishability under adaptive chosen-ciphertext attack (IND-CCA2) security. The IND-CCA2 analysis and primal/dual attack resistance of the proposed D-NTRU KEM are examined in detail. A comparison with similar protocols is provided regarding parameters, public/secret keys, and ciphertext sizes. The proposed scheme presents arithmetic simplicity and IND-CCA2 security that does not require any padding mechanism.

# INTRODUCTION

Public-key cryptosystems (PKC) are commonly used for essential purposes such as key sharing, authentication, and data encryption. Today, Diffie-Hellman (DH) key exchange (KE) (*Diffie & Hellman, 1976*) and RSA encryption/key encapsulation mechanism (KEM) (*Rivest, Shamir & Adleman, 1978*) are some of the widely used PKC. Shor's algorithm (*Shor, 1994*) proposed a polynomial-time solution to some computationally hard problems that guarantee the security assumptions of traditional PKC. It provides a solution to integer factorization (IF) and discrete logarithm problems (DLP) in polynomial time on a sufficiently large quantum computer. The requirement for constructing post-quantum secure PKC has emerged. While it will take time to build large-scale quantum computers, many initiatives exist to obtain post-quantum secure communication. In 2016, NIST started a standardization process (*National Institute of Standards and Technology (NIST), 2023*) to determine the standard PKC for the post-quantum era. One of the post-quantum secure cryptosystem families is the lattice-based constructions that provides worst-case assumptions, strong security guarantees, and relatively efficient applications. The lattice-based Number Theory Research Unit (NTRU) encryption scheme was proposed by

Corresponding author
Sedat Akleylek, akleylek@gmail.com

Hoffstein-Pipher-Silverman (*Hoffstein, Pipher & Silverman, 1998*). The security of the NTRU is based on the shortest vector problem. It provides relatively short and easy to construct keys and needs low memory with a high-speed guarantee. Many NTRU-like protocols have been proposed for today and post-quantum secure communication. Some current literature based on the NTRU is examined as follows.

The non-commutative ring structure was used in the MaTRU cryptosystem (*Coglianese & Goi, 2005*). The linear transformation of MaTRU provided significant speed improvements compared to the NTRU. The comparison analysis shows that MaTRU has a larger public key size than the NTRU, while the secret key size is smaller. In *Stehlé & Steinfeld (2011)*, the secret polynomial distribution of the NTRU was changed. The obtained construction provided efficient and flexible cryptographic structures. The hardness assumption of the proposed scheme was based on the ring learning with errors (RLWE) problem. The chosen-plaintext attack (CPA) resistance of this protocol was also presented. In 2013, ETRU encryption scheme was proposed by changing the algebraic structure of the NTRU (*Jarvis & Nevins, 2015*). The ring structure of ETRU was obtained by using Eisenstein integers. It is faster than the NTRU since it has smaller key sizes. The security analysis showed that ETRU is secure against chosen ciphertext attacks (CCA). In *Karbasi & Atani (2015)*, ILNTRU was constructed as a modified version of ETRU. The hardness assumption was based on the ring short integer solution (RSIS) and RLWE problems. It also provided indistinguishability under CPA (IND-CPA) resistance. In 2018, IND-CPA secure D-NTRU scheme was proposed in *Wang, Lei & Hu (2018)*. The security of the D-NTRU was explained based on the one-way assumption of the double-encrypted NTRU version. According to the comparison results, the D-NTRU was asymptotically faster than the NTRU. Many NTRU-based cryptosystems were also proposed in the NIST's standardization project and none was selected as a standard (*National Institute of Standards and Technology (NIST), 2023*). One of the NTRU-based cryptosystems is *Chen et al. (2022)*. It was designed as a KEM that guarantees IND-CCA2 security in the random oracle model (ROM). Applying the general transformation to the deterministic PKC provided a simple, fast, and compact structure.

In the literature, two methods were generally used to construct IND-CCA2 secure KEM: NAEP transformation (*Howgrave-Graham et al., 2003*) and Dent construction (*Dent, 2003*). A pseudo-trapdoor one-way function-based padding mechanism of NAEP provided the IND-CCA2 security (*Howgrave-Graham et al., 2003*). Dent's generic KEM construction contains special hash functions to obtain key derivation and masking data based on a one-way CPA secure encryption scheme (*Dent, 2003*). In this article, we follow the idea of Dent to obtain IND-CCA2 secure KEM as it is simple and does not require any padding mechanism. In addition, the D-NTRU encryption (*Wang, Lei & Hu, 2018*) was constructed using one-way hash functions that provide one-wayness against CPA security. So, the adaptation of Dent's approach will be more suitable and simple than NAEP for producing the D-NTRU-based IND-CCA2 secure KEM.

## Motivation and contribution

The design of post-quantum secure KEM is one of the significant open problems in the literature. The main aim of this article is to provide an IND-CCA2 secure KEM for this requirement. The proposed KEM is obtained following the D-NTRU encryption (*Wang, Lei & Hu, 2018*) and Dent's construction (*Dent, 2003*). The contributions of this article are summarized as follows:

– This is the first IND-CCA2 secure D-NTRU-based KEM scheme constructed with a one-way encryption function.
– The security analysis of the proposed KEM is given in the ROM. To provide IND-CCA2 security, the hybrid version of *Dent (2003)* and *Shoup (2001)* constructions are adapted.
– The proposed D-NTRU KEM is a solution to the IND-CCA2 security of the D-NTRU-based encryption specified as an open problem in *Wang, Lei & Hu (2018)*.
– The constructed KEM provides IND-CCA2 security without any padding mechanism or complex arithmetic operations.
– According to the proposed parameter set, a comparison with similar protocols is also presented.

## Organization

The rest of this article is organized as follows: In Section 2, some basic definitions and assumptions are recalled. In Section 3, the proposed D-NTRU-based KEM scheme and its correctness analysis are given. The security analysis against IND-CCA2 and primal/dual attacks are presented in Section 4. The comparisons are given in Section 5. Finally, Section 6 clarifies the conclusions.

## MATHEMATICAL BACKGROUND

The notations are summarized in Table 1.

In 2018, an NTRU variant double NTRU (D-NTRU) scheme was proposed in *Wang, Lei & Hu (2018)*. To explain the main properties of the D-NTRU, the composite NTRU (C-NTRU) was also defined. The hardness assumptions of the C-NTRU and the D-NTRU were based on the traditional NTRU scheme. Since this article aims to obtain the IND-CCA2 version of the D-NTRU, the main properties of the C-NTRU and the D-NTRU are recalled in the following.

In *Wang, Lei & Hu (2018)*, the C-NTRU was defined to explain the idea of the D-NTRU scheme. Let $p = 3$, $q_2 \in \mathbb{Z}^+$, $N$, and $q_1 \approx q$ be prime numbers such that $\gcd(q_1, q_2) = \gcd(q_1, p) = 1$. The C-NTRU scheme is recalled in Fig. 1.

In Fig. 1, the composite integers are used as moduli to obtain the public key. In step 7, $h[i] \mod q_1 q_2$ is computed with the Chinese remainder theorem (CRT), where $h[i] \equiv h_1[i] \mod q_1$ and $h[i] \equiv h_2[i] \mod q_2$ for $i \in [N]$. By following the steps of the key generation function, $h$ and $(f, f_p^{-1})$ are generated as the public and secret keys of the C-NTRU, respectively. The ciphertext of the C-NTRU $c = F_c(\phi, m) = \phi \otimes h + m$

**Table 1 Notations.**

| | | |
|---|---|---|
| $N$ | : | Dimension |
| $R_q = \frac{\mathbb{Z}_q[x]}{x^N-1}$ | : | Polynomial ring. |
| $q, q_1, q_2$ | : | Modulo values. |
| $\oplus$ | : | XOR operation. |
| $\otimes$ | : | Multiplication in a polynomial ring. |
| $x \leftarrow^r X$ | : | $x$ is chosen uniformly random from distribution $X$. |
| $\|x\|_\infty$ | : | For $x \in R$, $\|x\|_\infty = \max_{1 \le i \le N}\{x_i\} - \min_{1 \le i \le N}\{x_i\}$. |
| $\|x\|_2$ | : | For $x \in R$, $\|x\|_2 = (\sum_{i=1}^N (x_i - \bar{x})^2)^{1/2}$, where $\bar{x} = \frac{1}{N}\sum_{i=1}^N x_i$. |
| $\kappa \in \mathbb{Z}_{\ge 0}$ | : | The main security parameter. |
| $\perp$ | : | Error message. |
| $d \in \mathbb{Z}^+$ | : | The parameter of polynomial spaces. |
| | | For D-NTRU, $d = d_f = d_g$. |
| $\delta, \phi$ | : | The failure parameters. |
| $x_p^{-1}$ | : | The inverse of the polynomial x in mod p. |
| | | Let $x = f$. Then, $f \otimes f_p^{-1} \equiv 1 \mod p$. |
| $\neg Z$ | : | The complement of $Z$. |
| $L(d, d)$ | : | Ternary polynomials. If $x \leftarrow^r L(d, d)$, then $d$ coefficient of $x$ is equal to 1, $d$ coefficient of $x$ is equal to $-1$, and the others are equal to 0. |
| Kdf | : | Key derivation function such that $R_{q_2} \to \{0, 1\}^N$. |
| H | : | Hash function such that $R_{q_1} \to \{0, 1\}^N$. |

mod $q_1q_2$ is obtained by running the encryption procedure with input message $m \in R_p$. In the decryption phase, $c$ is decrypted to $m$ with the help of the private key $f_p^{-1}$.

**Remark 1** *If $q_1 = q$ and $q_2 = 1$, then the C-NTRU$\equiv$NTRU (Wang, Lei & Hu, 2018).*

Based on the C-NTRU encryption, the C-NTRU one-way problem was defined in *Wang, Lei & Hu (2018)*.

**Definition 1 (The C-NTRU One-Way Problem (*Wang, Lei & Hu, 2018*)).** *Let $h(X) = \sum_{i \in [N]} h[i]X^i \in R_{q_1q_2}$ be the public key and $c = F_c(\phi, m) = \phi \otimes h + m \in R_{q_1q_2}$ be the ciphertext of the C-NTRU scheme, where $(\phi, m) \in L(d, d) \times R_p$. The main purpose is to find another polynomial pair $(\phi, m)$ under the C-NTRU function $F_c$ that produces the ciphertext c.*

The C-NTRU one-way problem was obtained by following the NTRU one-way problem (*Wang, Lei & Hu, 2018*).

**Definition 2 (The NTRU One-Way Problem (*Hoffstein, Pipher & Silverman, 1998*)).** *Let $h$ be the public key and $c = F(\phi, m) = h \otimes \phi + m \in R_q$ be the ciphertext of the NTRU scheme, where $(\phi, m) \in L(d, d) \times R_p$. The main purpose is to find another polynomial pair $(\phi, m)$ under the NTRU encryption function $F$ that produces the ciphertext c.*

The hardness assumption of the NTRU one-way problem is recalled in Definition 3.

**Definition 3 (The Hardness Assumption of NTRU One-Way Problem).** *Any probabilistic polynomial time (PPT) algorithm that solves the NTRU one-way problem is*

```
 1:  function KEY GENERATION(1^κ)
 2:      f ←^r L(d_f, d_f − 1)                                      ▷ sk
 3:      f_p^{-1}                                                   ▷ sk
 4:      g ←^r L(d_g, d_g − 1)
 5:      h_1 = p ⊗ f_{q_1}^{-1} ⊗ g   mod q_1                      ▷ h_1 ∈ R_{q_1}
 6:      h_2 ←^r R_{q_2}
 7:      h(X) = Σ_{i∈[N]} h[i]X^i   mod q_1 q_2                    ▷ pk: h(X) ∈ R_{q_1 q_2}
 8:  end function
 9:  function ENCRYPTION(pk)
10:      m ∈ R_p
11:      φ ←^r L(d, d)
12:      c = F_c(φ, m) = φ ⊗ h + m   mod q_1 q_2                   ▷ ct: c ∈ R_{q_1 q_2}
13:  end function
14:  function DECRYPTION(c)
15:      s = f ⊗ c   mod q_1                                       ▷ s ∈ R_{q_1}
16:      m = s ⊗ f_p^{-1}   mod p                                  ▷ m ∈ R_p
17:  end function
```
sk: Secret key, pk: Public key, ct: Ciphertext

**Figure 1  Algorithm 1: C-NTRU encryption *Wang, Lei & Hu (2018)*.**

negligible for $\kappa$. In other words, for sufficiently large $\kappa$, it is impossible to develop a PPT algorithm to solve the NTRU one-way problem with a non-negligible probability (*Howgrave-Graham et al., 2003*).

The relation between the NTRU and the C-NTRU one-way problems is given by Fact 1.
**Fact 1** *Let $q_1 > \delta$. Then, the C-NTRU one-way problem is reduced to the NTRU one-way problem in polynomial time (*Wang, Lei & Hu, 2018*, Theorem 4).*

The C-NTRU ciphertext distribution problem is described in Definition 4.
**Definition 4 (The C-NTRU Ciphertext Distribution Problem (*Wang, Lei & Hu, 2018*)).**
*Let $h(X) = \sum_{i\in[N]} h[i]X^i \in R_{q_1 q_2}$ be the public key of the C-NTRU scheme. The main purpose is to distinguish the distributions of uniformly random chosen ciphertext $c \leftarrow^r R_{q_1 q_2}$ and $c = F_c(\phi, m)|(\phi, m) \leftarrow^r L(d, d) \times R_p$ that is produced using the C-NTRU ciphertext function $F_c$.*

The relationship between the C-NTRU one-way problem and the ciphertext distribution is summarized Fact 2.
**Fact 2** *If $q_1 > \delta + 2$, the C-NTRU ciphertext distribution problem is reduced to the C-NTRU one-way in polynomial time (*Wang, Lei & Hu, 2018*, Theorem 4).*

By showing reductions to the C-NTRU properties, the D-NTRU scheme was also constructed to obtain more efficient the NTRU-based public-key encryption (*Wang, Lei & Hu, 2018*). In the proposed scheme, double encryption is provided with the usage of twin primes. The main structure of the D-NTRU is remembered in Definition 5.
**Definition 5 (The D-NTRU (*Wang, Lei & Hu, 2018*)).** *Let $p = 3, N$, and $q_1 + 2 = q_2 \approx q$. Then, the D-NTRU encryption scheme, using $q_1$ and $q_2$ twin primes, is given in Fig. 2.*

The proposed D-NTRU was designed as a double version of the NTRU that provide one-time pad encryption. The ct component $c_1$ allows some parameters, such as $r_1$ and $r_2$,

```
 1: function KEY GENERATION(1^κ)
 2:     f ←^r L(d_f, d_f − 1)                                      ▷ sk
 3:     f_p^{-1}                                                   ▷ sk
 4:     g ←^r L(d_g, d_g − 1)
 5:     h_1 = p ⊗ f_{q_1}^{-1} ⊗ g   mod q_1                      ▷ pk: h_1 ∈ R_{q_1}
 6:     h_2 ←^r R_{q_2}                                           ▷ pk
 7:     G = p^{-1} ⊗ g_{q_1}^{-1}   mod q_1                       ▷ sk: G ∈ R_{q_1}
 8: end function
 9: function ENCRYPTION(pk)
10:     M ∈ R_{q_2}
11:     r_1 ←^r L(d, d)
12:     r_2 ←^r R_p
13:     c_1 = r_1 ⊗ h_1 + r_2   mod q_1                           ▷ c_1 ∈ R_{q_1}
14:     c_2 = r_1 ⊗ h_2 + r_2 + M   mod q_2                       ▷ c_2 ∈ R_{q_2}
15:     c = (c_1, c_2)                                            ▷ ct
16: end function
17: function DECRYPTION(c)
18:     s = f ⊗ c_1   mod q_1                                     ▷ s ∈ R_{q_1}
19:     r_2 = s ⊗ f_p^{-1}   mod p                                ▷ r_2 ∈ R_p
20:     r_1 = G ⊗ (s − f ⊗ r_2)   mod q_1                         ▷ r_1 ∈ R_{q_1}
21:     M = c_2 − r_1 ⊗ h_2 − r_2   mod q_2                       ▷ M ∈ R_{q_2}
22: end function
```

sk: Secret key, pk: Public key, ct: Ciphertext

**Figure 2 Algorithm 2: D-NTRU encryption scheme.**

to be shared and recovered, while $c_2$ provides one-time pad-like encryption. The D-NTRU encryption scheme using $q_1$ and $q_2$ twin primes is given in Fig. 2.

The relation between secret polynomials and components of the D-NTRU is recalled in Corollary 1.

**Corollary 1** *Let $f$ and $g$ be secret polynomials of the D-NTRU. If $||f_{q_1}^{-1} \otimes g \quad \mod q_1||_\infty > 2$ and $||f \otimes g_{q_1}^{-1} \quad \mod q_1||_\infty > 2$, the decryption of the D-NTRU scheme will not fail (Wang, Lei & Hu, 2018, Fact 1).*

The conditions to prevent possible errors during the decryption phase of the D-NTRU are explained with Theorem 1.

**Theorem 1** *Let $\delta = 2(p \min\{2d_g − 1, 2d\} + 2d_f − 1)$. If $q_1 > \delta$, there is no decryption failure in the D-NTRU (Wang, Lei & Hu, 2018, Theorem 1).*

**Proof 1** *In the D-NTRU algorithm, the decryption parameter $s = f \otimes c_1 \quad \mod q_1 = f \otimes (r_1 \otimes h_1 + r_2) \quad \mod q_1 = f \otimes (r_1 \otimes (p \otimes f_{q_1}^{-1} \otimes g) + r_2) \quad \mod q_1 = r_1 \otimes p \otimes g + f \otimes r_2 \quad \mod q_1$ must be computed correctly to recover the message. Since $q_1 > \delta$, Eq. (1) is satisfied.*

$$||s||_\infty = || \overbrace{r_1}^{\in L(d,d)} \otimes p \otimes \overbrace{g}^{\in L(d_g, d_g − 1)} + \overbrace{f}^{\in L(d_f, d_f − 1)} \otimes \overbrace{r_2}^{\in R_p} ||_\infty \qquad (1)$$
$$\leq p \min\{2d_g − 1, 2d\} + \min\{||r_2||, 2d_f − 1\}$$

Based on Theorem 1, the relation between the ciphertext and the parameters of the D-NTRU is explained with Theorem 2. Please see the detailed proof of Theorem 2 in Wang, Lei & Hu (2018).

**Theorem 2.** *Let $\delta = 2(p \min\{2d_g - 1, 2d\} + 2d_f - 1)$ and $q_1 > \delta$. Then, there is at most one $(r_1, r_2) \in L(d, d) \times R_p$ pair that satisfies $c_1 = h_1 \otimes r_1 + r_2 \mod q_1$ for any $c_1 \in R_{q_1}$* (*Wang, Lei & Hu, 2018*).

The invalid ciphertext of the D-NTRU is examined in Theorem 3. Please see the detailed proof of Theorem 3 in *Wang, Lei & Hu (2018)*.

**Theorem 3.** *Let $q_1 > \delta + 2$ and $c = (c_1, c_2) \in R_{q_1} \times R_{q_2}$ be the encrypted version of the message M for $(r_1, r_2) \leftarrow^r L(d, d) \times R_p$. For any $i \in [N]$, $(c_1 + X^i \mod q_1, c_2 + X^i \mod q_1)$ and $(c_1 - X^i \mod q_1, c_2 - X^i \mod q_1)$ are the invalid ciphertexts of the D-NTRU cryptosystem, where $r_2[i] = 1$ and $r_2[i] = -1$, respectively* (*Wang, Lei & Hu, 2018*).

The distribution problem of the D-NTRU is recalled in Definition 6.

**Definition 6 (The Distribution Problem of the D-NTRU (*Wang, Lei & Hu, 2018*)).** *Let $(h_1, h_2)$ be the public key pair of the D-NTRU scheme. The main purpose is to distinguish the distribution of uniformly random chosen ciphertext $(s_1, s_2) \leftarrow^r R_{q_1} \times R_{q_2}$ and the distribution of the D-NTRU ciphertext function*
*$(s_1 = h_1 \otimes r_1 + r_2 \in R_{q_1}, s_2 = h_2 \otimes r_1 + r_2 \in R_{q_2}) | (r_1, r_2) \leftarrow^r L(d, d) \times R_p$.*

The relation between Definitions 4 and 6 is summarized with Fact 3.

**Fact 3** *The D-NTRU distribution problem is reduced to the C-NTRU ciphertext distribution problem in polynomial time* (*Wang, Lei & Hu, 2018*, Theorem 6).

The one-way property of the D-NTRU scheme is explained in Corollary 2.

**Corollary 2** *Let the D-NTRU distribution problem is reduced to the C-NTRU ciphertext distribution problem in polynomial time. Then, since the C-NTRU scheme provides the one-way property, the D-NTRU scheme also has the one-way property* (*Wang, Lei & Hu, 2018*).

The main properties of the D-NTRU encryption scheme are defined by explaining its relationship with the C-NTRU and the NTRU. In this section, the relations between hard problems and their main properties are expressed to show the one-wayness property of the D-NTRU encryption. Based on Corollary 2 and Dent's KEM construction (*Dent, 2003*), the proposed KEM is explained in Section 3.

## PROPOSED SCHEME

In this section, the proposed D-NTRU based IND-CCA2 secure KEM is detailed. Due to the one-way structure of the D-NTRU encryption function, to obtain IND-CCA2 security, Dent's one-way KEM construction (*Dent, 2003*) is added. The basic idea of obtaining the D-NTRU-based IND-CCA2 KEM is based on the modified D-NTRU encryption and Dent's KEM components. The proposed IND-CCA2 secure D-NTRU-based KEM scheme is given in Fig. 3.

In Fig. 3, the public and secret keys are generated using the key generation procedure of Algorithm 3. To construct IND-CCA2 secure KEM, Dent's KEM design idea, based on a one-way function, is used. In the encapsulation procedure of Algorithm 3, the ct components of Algorithm 2 are reevaluated in the following way.

- $c_1 = r_1 \otimes h_1 + r_2 \mod q_1$ is modified as
  $c_1 = (r_1 + r_2) \otimes h_1 + (p \otimes g + 1) \otimes r_2 \mod q_1$ to prevent a IND-CCA2 based attack that is explained in Remark 3.

```
 1: function KEY GENERATION(1^κ)
 2:     f ←^r L(d_f, d_f − 1), f_p^{−1}                              ▷ sk
 3:     g ←^r L(d_g, d_g − 1)
 4:     f_{q_1}^{−1}, g_{q_1}^{−1}, p^{−1}
 5:     h_1 = p ⊗ f_{q_1}^{−1} ⊗ g mod q_1                          ▷ pk: h_1 ∈ R_{q_1}
 6:     h_2 ←^r R_{q_2}                                             ▷ pk
 7:     G = p^{−1} ⊗ g_{q_1}^{−1} mod q_1                           ▷ sk: G ∈ R_{q_1}
 8:     (h_1, h_2)                                                  ▷ pk
 9:     (f, f_p^{−1}, G)                                            ▷ sk
10: end function
11: function ENCAPSULATION(pk)
12:     M ∈ R_{q_2}
13:     r_1 ←^r L(d, d)
14:     r_2 ←^r R_p
15:     c_1 = (r_1 + r_2) ⊗ h_1 + (p ⊗ g + 1) ⊗ r_2 mod q_1        ▷ ct: c_1 ∈ R_{q_1}
16:     c_2 = r_1 ⊗ h_2 + r_2 + (M ⊕ H(c_1)) mod q_2               ▷ ct: c_2 ∈ R_{q_2}
17:     C = (c_1, c_2)                                              ▷ ct
18:     K = kdf(M)
19:     return (K, C)                                               ▷ ssk
20: end function
21: function DECAPSULATION((C, sk))
22:     C = (c_1, c_2)
23:     r_2 = (f ⊗ c_1 mod q_1) ⊗ f_p^{−1} mod p                   ▷ r_2 ∈ R_p
24:     r_1 = [G ⊗ (f ⊗ c_1 − f ⊗ r_2] − f ⊗ r_2 − r_2 mod q_1     ▷ r_1 ∈ R_{q_1}
25:     M' = (c_2 − (r_1 ⊗ h_2 + r_2)) ⊕ H(c_1) mod q_2
26:     K = kdf(M')
27:     return (K)                                                  ▷ ssk
28: end function
```

sk: Secret key, pk: Public key, ct: Cipher-text, ssk: Secret shared key

**Figure 3 Algorithm 3: D-NTRU-based IND-CCA2 secure KEM scheme.**

- $c_2 = r_1 \otimes h_2 + r_2 + M \quad \mathrm{mod}\, q_2$ is changed as
  $c_2 = r_1 \otimes h_2 + r_2 + (M \oplus \mathrm{H}(c_1)) \quad \mathrm{mod}\, q_2$ to provide one of IND-CCA2 components of *Dent (2003)*.

The encapsulation steps are completed by computing the shared key using Kdf. To decapsulate $C$, $r_1$ and $r_2$ are recovered using secret keys $(f, f_p^{-1}, G)$. Then, the recovered message $M'$ is used to obtain shared key $K$ under Kdf.

In Fig. 3, Kdf and H functions are used to ensure IND-CCA2 security (*Dent, 2003*; *Shoup, 2001*).

1. Kdf: It is a cryptographic algorithm that derives one or more secret keys from various values using a pseudo-random function. It is modeled as ROM based on hash function properties. In the proposed KEM, *Shoup (2001)*'s Kdf1 function ($R_{q_2} \rightarrow \{0, 1\}^N$) is chosen as Kdf.

2. H: $R_{q_1} \rightarrow \{0, 1\}^N$: It is a hash function that provides entropy smoothing regarding security properties. For high-entropy input ciphertext byte sequences, the output is computationally indistinguishable from a random byte sequence with the same length.

In the proposed scheme, SHA series hash function family and *National Institute of Standards and Technology (NIST) (2022)* and *Shoup (2001)*'s Kdf1 functions can be used depending on the choice of $\kappa$.

## Correctness

The equality of keys obtained by encapsulation and decapsulation is examined in the correctness analysis of the D-NTRU KEM. In Fig. 3, if step 23 does not work correctly, the decapsulation failure consists. So, the parameters should be chosen according to Theorem 4.

**Theorem 4.** If $q_1 > \phi = 2(p\min\{2d, 2d_g - 1\} + p(2d_g - 1) + p\min\{2d_f - 1, 2d_g - 1\} + 2d_f - 1)$, the decapsulation failure does not occur in the proposed D-NTRU KEM.

**Proof 2.** If the first component of the step 23 in *Fig. 3* is rewritten, *Eq. (2)* is derived.

$$
\overbrace{\|f \otimes c_1\|_\infty}^{\in R_{q_1}} \leq \|(\overbrace{r_1}^{L(d,d)} \otimes p \otimes \overbrace{g}^{L(d_g, d_g-1)} + \otimes \overbrace{r_2}^{R_p} \otimes p \otimes g +
$$

$$
\overbrace{f}^{L(d_f, d_f-1)} \otimes p \otimes g \otimes r_2 + f \otimes r_2)\|_\infty \tag{2}
$$

$$
\leq p\min\{2d, 2d_g - 1\} + p\min\{\|r_2\|, 2d_g - 1\} +
$$

$$
p\min\{2d_f - 1, \|r_2\|, 2d_g - 1\} + \min\{\|r_2\|, 2d_f - 1\}
$$

$$
\leq p\min\{2d, 2d_g - 1\} + p(2d_g - 1) + p\min\{2d_f - 1, 2d_g - 1\} + 2d_f - 1
$$

*Based on Theorems 1 and 2, if $q_1 > \phi$ is satisfied in the parameter selection, there will be no problem in the correctness.*

Let $C = (c_1, c_2)$ is rewritten with Eq. (3) to show the correctness of the D-NTRU KEM.

$$
c_1 = (r_1 + r_2) \otimes h_1 + (p \otimes g + 1) \otimes r_2 \quad \mod q_1
$$
$$
c_2 = r_1 \otimes h_2 + r_2 + (M \oplus \mathrm{H}(c_1)) \quad \mod q_2 \tag{3}
$$

By using Eq. (3), to recover the message $M$, the running process of the decapsulation procedure is explained with Eqs. (4) and (5). If the step 23 of Fig. 3 is rewritten, Eq. (4) is obtained.

$$
= (f \otimes c_1 \quad \mod q_1) \otimes f_p^{-1} \quad \mod p
$$

$$
= (f \otimes ((r_1 + r_2) \otimes h_1 + (p \otimes g + 1) \otimes r_2) \quad \mod q_1) \otimes f_p^{-1} \quad \mod p
$$

$$
= \overbrace{(f \otimes r_1 \otimes p \otimes f_{q_1}^{-1} \otimes g + f \otimes r_2 \otimes p \otimes f_{q_1}^{-1} \otimes g + f \otimes p \otimes g \otimes r_2 + f \otimes r_2)}^{f \otimes f_{q_1}^{-1} \equiv 1 \quad \mod q_1, \ r_{i=\{1,2\}} \otimes p \otimes g \quad \mod p \equiv 0} \quad \mod q_1 \tag{4}
$$

$$
\otimes f_p^{-1} \quad \mod p
$$

$$
= r_2 \quad \mod p
$$

To decapsulate $c_1$ and $c_2$, the step 24 of Fig. 3 is reevaluated with Eq. (5).

$$= [G \otimes ( \overbrace{f \otimes c_1}^{r_1 \otimes p \otimes g + r_2 \otimes p \otimes g + f \otimes p \otimes g \otimes r_2 + f \otimes r_2} \quad -f \otimes r_2)] - f \otimes r_2 - r_2 \mod q_1$$

$$= [p^{-1} \otimes g_{q_1}^{-1} \otimes (r_1 \otimes p \otimes g + r_2 \otimes p \otimes g + f \otimes p \otimes g \otimes r_2 + f \otimes r_2 - f \otimes r_2)]$$
$$- f \otimes r_2 - r_2 \mod q_1$$

$$= [r_1 + r_2 + f \otimes r_2] - f \otimes r_2 - r_2 \mod q_1 \tag{5}$$

$$= r_1 \mod q_1$$

$$M' = (c_2 - (r_1 \otimes h_2 + r_2)) \oplus H(c_1) \mod q_2$$
$$= ((r_1 \otimes h_2 + r_2 + (M \oplus H(c_1))) - (r_1 \otimes h_2 + r_2)) \oplus H(c_1) \mod q_2$$

$$M' \equiv M \mod q_2$$

The relationship between modulo $q_1$ and distribution parameter $d$ is explained in Corollary 3.

**Corollary 3** *Let $d = d_f = d_g$ and $p = 3$ (Wang, Lei & Hu, 2018). Based on Theorems 3 and 4, since $q_1 > \phi + 2$, $q_1 > \phi + 2 = 2(p \min\{2d, 2d_g - 1\} + p(2d_g - 1) + p \min\{2d_f - 1, 2d_g - 1\} + 2d_f - 1) + 2 = 40d - 18$ is obtained. To prevent decapsulation failures and invalid ciphertext, $d = \lfloor \frac{q_1 + 18}{40} \rfloor$ must be satisfied. Then, $M' = M$.*

## SECURITY ANALYSIS OF D-NTRU KEM

In this section, the IND-CCA2 security analysis of the proposed KEM and its resistance to some lattice-based attacks are examined.

### The IND-CCA2 security of the proposed KEM

In the security analysis, the idea of IND-CCA2 secure KEM from Dent's one-way IND-CPA secure encryption scheme (Dent, 2003) is followed. The model of IND-CCA2 security is constructed by adapting (Bogdanov, 2005; Dent, 2003; Shoup, 2001) to the D-NTRU problem. The attacker's behaviors in the IND-CCA2 security are examined based on the game-based security analysis.

In this model, an Attacker (A), modeled as a PPT Turing machine, has the authority to run all algorithms and can obtain all communication-related media. *A* can also access the decapsulation oracle to decapsulate any capsulated pair. According to Dent's KEM structure, the proposed KEM is secure unless *A* has a significant advantage over the Game$_1$ against a mythical challenger.

Game$_1$: There are three consecutive operations, such as start, challenge, and result, in the Game$_1$. *A* aims to gain an advantage in the basic IND game by performing these operations. The visualization sub-steps of Game$_1$ is given in Fig. 4. Figure 4 shows the parameters obtained during the Game$_1$ based on the action of *A*. The summarized reactions of Fig. 4 are defined as follows.

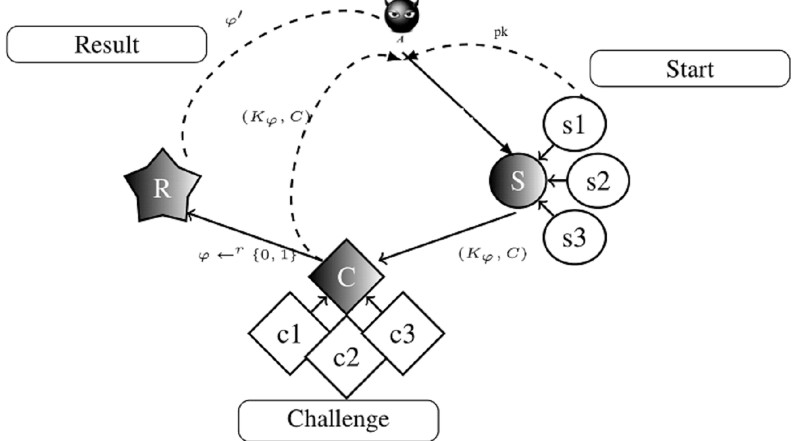

**Figure 4  The basic IND security game steps.**     

- **Start:** There are three sub-steps.

s1: Based on the security parameter $\kappa$, the key pair $(pk, sk)$ is generated by the challenger. $pk$ is sent to $A$ and $sk$ is held by the challenger.

s2: $A$ runs until the challenger receives the capsulated key pair. Then, it queries the decapsulation oracle to find the key that is associated with the capsulated key. It is ready to take on a challenge when $A$ made enough queries.

s3: The following steps are taken by the $A$ when generating the encapsulated key pair for the challenger. $A$ submits two different capsulated keys:

  ★ $(K_0, C) = \text{Encapsulation}(pk)$
  ★ $K_1 \leftarrow^r \{0, 1\}^N$

- **Challenge:** It is completed when the following sub-steps are done.

c1: The challenger chooses bit $\varphi \leftarrow^r \{0, 1\}$ and sends the capsulated key $(K_\varphi, C)$ to $A$.

c2: $A$ can perform any number of additional capsulation and computations. According to the number of queries that $A$ can do, the obtained security properties are defined as follows.

  ★ In the non-adaptive IND-CCA, $A$ cannot make further requests to the decapsulation oracle before estimating $\varphi$.

  ★ In IND-CCA2, $A$ can make further requests to the decapsulation oracle before the prediction, while the challenger ciphertext $C$ cannot be submitted.

c3: $A$ works until he/she generates the guess bit $\varphi'$. Then, $A$ queries the decapsulation oracle to find the ciphertext $C$, which is associated with the capsulated key pair.

- **Result:**

1. If $\varphi' = \varphi$, A wins the game.
2. Let $p$ be the advantage of A in Game$_0$. In Eq. (6), if $p$ is a negligible function of $\kappa$, then the D-NTRU KEM is said to be IND-CCA2 secure.

$$p = Pr[\varphi = \varphi'] - 1/2 \tag{6}$$

Let's prove that Eq. (6) is negligible in the proposed KEM with Game$_2$.

Game$_2$: In the IND-CCA2 KEM scheme, A can query the decapsulation oracle more than once. Following the idea of Theorem 4 in Dent (2003), the IND-CCA2 analysis of the D-NTRU KEM is examined with Theorem 5. Let;

- $|H|$ be the output length of $H : R_{q_1} \to \{0,1\}^N$.
- $T$ be the execution time of encryption process.
- $|M|$ be the space size of $M \in R_{q_2}$.
- $Q_D, Q_H$, and $Q_K$ be the maximum number of the decapsulation, hash and Kdf oracles queries in the ROM, respectively.

**Theorem 5** *Let the D-NTRU-(Key Generation, Encryption, Decryption), given in Fig. 2, be a one-way encryption scheme and the D-NTRU KEM be the KEM obtained from this encryption by following Dent's construction. Suppose that there is an A in the ROM that can break the IND-CCA2 security with probability p in time t. Then, there is also an algorithm that inverts the underlying one-way encryption function with probability $p' \geq p - \frac{Q_D}{2^{|H|}} - \frac{Q_D}{|M|}$ and in time $t' \leq t + (Q_H + Q_D + Q_K)T$.*

**Proof 3** *It is shown that if there is an A that breaks the proposed scheme with a non-negligible probability, there will be an algorithm that reverses the underlying one-way encryption scheme with a non-negligible probability. Note that it is assumed that A can query the oracle at any time in the IND-CCA2 security. The following changes are done in Game$_2$.*

- *The challenger selects the challenge key pair $(K_\varphi, C)$ at the beginning.*

*– If A queries the decapsulation oracle with input $C = (C_1, C_2)$, it produces the error term $\perp$ at any time. The only difference compared with Game$_0$ is that A queries the decapsulation oracle with ciphertext C before obtaining $(K_\varphi, C)$. To analyze the effects of this difference on the advantage of A, adapted Lemmas 1 and 2 are used.*

**Lemma 1** *Let $Y_1$, $Y_2$ and $Z$ be an events such that $Pr[Y_1|\neg Z] = Pr[Y_2|\neg Z]$. Then,*

$$|\Pr[Y_1] - \Pr[Y_2]| \leq \Pr[Z] = \frac{Q_D}{|M|} + \frac{Q_D}{2^{|H|}}, \tag{7}$$

*where $|M|$ and $|H|$ be the message space size and the output length of H, respectively.*
**Proof** *Let $Y_1$ and $Y_2$ be the events that A wins in Game$_1$ and Game$_2$, respectively. Note that if A wins the games, he/she can correctly guess $\varphi$ from $(K_\varphi, C)$, where $(K_\varphi, C)$ is initially selected at just $Y_2$. Let Z be the event that A asks to decapsulation of C before the challenge is*

1: Initially, kdfList and decList lists are created and reset to zero.
2: pk=$(h_1, h_2)$ and challenger ciphertext $C^* = (C_1^*, C_2^*)$ are received from the system to invert. pk is sent to $A$ while $C^*$ is not sent to $A$ yet.
3: $A$ is allowed to run until it requests an encapsulated key pair query. During this time, two situations can occur.

  a. If $A$ queries the decapsulated value of $C = (C_1, C_2)$, the following steps are executed.

   i. If $C(C_1, C_2) = (C_1^*, C_2^*)$, operations are canceled with the output $\perp$.
   ii. It is checked whether there is a $(K, C)$ pair in the declist. If $(K, C) \in$ decList, operations are canceled with the output $K$.
   iii. If $(K, C) \notin$ decList, then $K \leftarrow^r \{0,1\}^N$ is chosen. $(K, C)$ is constructed and added to the decList.
   iv. The output is $K$.

  b. If $A$ queries the evaluation of any input $X$ in the kdf function, the following steps are executed.

   i. It is checked whether there is a $(K, X)$ pair in the kdfList. If $(K, X) \in$ kdfList, operations are canceled with the output $K$.
   ii. $K \leftarrow^r \{0,1\}^N$ is chosen. If $X \notin R_{q_2}$, then $(K, X)$ is constructed and added to the kdfList. The operations are canceled with the output $K$.
   iii. By using the $X$ input, $C = (c_1 = (r_1 + r_2) \otimes h_1 + (p \otimes g + 1) \otimes r_2 \mod q_1, c_2 = r_1 \otimes h_2 + r_2 + (X \oplus H(c_1)) \mod q_2)$ ciphertext is generated. If $C = C^*$, then the message $X$ that produces $C^*$ ciphertext. Since the purpose of the algorithm is obtained, the operations are terminated.
   iv. It is checked whether there is a $(K', C)$ pair in the decList. If $(K', C) \in$ decList, then $(K, X)$ is added to the kdfList. The operations are canceled with the output $K'$.
   v. Otherwise, the $(K, C)$ pair obtained in the Step b.iii. is added to the decList. The operations are canceled with the output $K$.

4: To generate the challenging key pair, $K^* \leftarrow^r \{0,1\}^N$ is chosen. The constructed $(C^*, K^*)$ is sent to $A$.
5: $A$ runs until it produces the guess $\varphi'$ as an output. Return to Step 3.
6: If the operations does not end, the output is obtained by choosing $X \leftarrow^r R_{q_2}$. In other words, $X$ is the inverse of $C^*$.

**Figure 5 Algorithm 4: Possible queries for the security analysis.**

done. If $Z$ does not occur, $A$ can only get the same information when querying oracles. Since, $Pr[Y_1|\neg Z] = Pr[Y_2|\neg Z]$, $|Pr[Y_1] - Pr[Y_2]| \leq Pr[Z]$ is obtained. In the IND-CCA2, the challenge ciphertext is chosen uniformly random from the possible ciphertext distribution. Since $c_1 = (r_1 + r_2) \otimes h_1 + (p \otimes g + 1) \otimes r_2 \mod q_1$ and $c_2 = r_1 \otimes h_2 + r_2 + (M \oplus H(c_1)) \mod q_2$, where $C = (c_1, c_2)$ in the proposed KEM, decapsulation and hash oracles are queried in the event $Z$.

The probability that $A$ guesses $C$ with an decapsulation oracle query is $\frac{1}{|M|}$ and hash oracle is $\frac{1}{|H|}$. Since $A$ can make multiple queries, $A$ can obtain decapsulation$(C, sk)$ and $C' = H(\text{decapsulation}(C, sk))$ with a total probability $\frac{Q_D}{|M|} + \frac{Q_D}{2^{|H|}}$, if event $Z$ occurs. So, $Y_2$ is only negligibly smaller than $Y_1$ since $\frac{1}{|M|}$ and $\frac{1}{|H|}$ are negligible in the one-way function.

**Lemma 2** Let $A$ has a non-negligible advantage in the $Game_1$. Then, there will also be an $A'$ with a non-negligible advantage in $Game_2$.

**Proof 2** Suppose that there is an $A'$ who breaks KEM with probability $p'$ in $Game_2$. He/She runs the algorithm, given in Fig. 5, to reverse the one-way D-NTRU function. At the end of Algorithm 4, the winning probability of $A'$ is computed. This value is equal to $A'$'s success in reversing the challenge ciphertext $C^*$. Let consider Step 3.b.iii in Fig. 5. The probability of

*obtaining the plaintext $X^*$ that creates the challenging ciphertext $C^*$ is equivalent to the probability that $A'$ wins $Game_2$. The derivation of $X^*$ includes operation steps based on Kdf and H functions. Let V be the event of querying the Kdf function with $X^* = decapsulation(C^*, sk)$ at any time. The probability of outputting $X^*$, which is the plaintext of $C^*$, is computed with Eq. (8).*

$$\Pr[X^*] = \Pr[V]\Pr[X^*|V] + \Pr[\neg V]\Pr[X^*|\neg V] \tag{8}$$

*when the event V does not occur, A cannot obtain anything about $kdf(X^*) = Decapsulation(C^*, sk)$. So, $\Pr[\neg V] = 0$. $\Pr[V] = \frac{Q_D}{|M|} + \frac{Q_D}{2^{|H|}}$ is obtained with Eq. (7). If Eq. (8) is rewritten, Eq. (9) is obtained.*

$$
\begin{aligned}
\Pr[X^*] &= \overbrace{\Pr[V]}^{\frac{Q_D}{|M|}+\frac{Q_D}{2^{|H|}}} \overbrace{\Pr[X^*|V]}^{\equiv 1} + \overbrace{\Pr[\neg V]}^{0} \Pr[X^*|\neg V] \\
&\leq \Pr[V] \\
&\leq \frac{Q_D}{|M|} + \frac{Q_D}{2^{|H|}}
\end{aligned}
\tag{9}
$$

*Since Kdf is modeled as a ROM and the D-NTRU provides one-wayness property, the total number of queries $\frac{Q_D}{|M|} + \frac{Q_D}{2^{|H|}}$ as a function of $\kappa$ is negligible. Since Eq. (6) is hold, there is no an algorithm that can reverse the given $C^*$ ciphertext is obtained with a negligible. So, the proposed D-NTRU-based KEM is IND-CCA2 secure.*

### Basic lattice-based attacks

The security of the NTRU-based protocols is related to the shortest vector problem (SVP). The primal and dual attacks can be carried out for the NTRU-like protocols, such as the D-NTRU encryption and KEM, to find the short vectors in a lattice. Therefore, the parameter set should be chosen so that it is impossible to find short vectors (*Elverdi, Akleylek & Kirlar, 2022*). The primal and dual attack resistance of the D-NTRU KEM is examined as follows.

- Primal attack aims to estimate the hardness of the learning with errors (LWE)-based crptosystems. By constructing an integer embedded lattice, it tries to solve the unique short vector problem (u-SVP). In other words, it reduces the LWE problem to the unique SVP by using the embedding technique. Then, it uses Block Korkin-Zolotarev (BKZ) lattice reduction to find the shortest vector. The hardness of the core-SVP estimates the complexity of the primal attack as $2^{0.3496b}$, where $b$ is the block size of the BKZ algorithm (*Liang et al., 2022*). So, in the primal attack resistance of the D-NTRU-KEM algorithm, the reduced base $V = (v_1, \ldots, v_d)$ is computed with the BKZ-b. In the BKZ-b algorithm, $b$ is selected as 364 independent of parameters for $n = 128$ bit security level in the local model (*Liang et al., 2022*; *Hoffstein, Pipher & Silverman, 1998*). Therefore, the core-SVP cost of primal attack is estimated as $\lfloor 0.3496 \times 364 = 127 \rfloor$ for

**Table 2 Computational basics for parameters.**

| Primitives | | | Properties and Components |
|---|---|---|---|
| Dimension | : | $N$ | Prime, $d \approx N/3$ |
| Mod | : | $q_1, q_2$ | Twin primes $q_1 + 2 = q_2$ |
| Polynomial coefficients | : | $d_f = d_g = d$ | $q_1 > 40d - 18$ |
| Message security | : | $\mathrm{ms} = \log_2 \sqrt{\frac{N!}{d!^2(N-2d)!}}$ bit | |
| Key security | : | $\mathrm{ks} = \log_2 \sqrt{\frac{N!}{d!(d-1)!(N-2d+1)!}}$ bit | |
| Security parameter | : | $\kappa = \lfloor \min\{ms, ks\} \rfloor$ | |
| Public key size | : | $\mathrm{pk} = \frac{N(\log_2 q_1 + \log_2 q_2)}{8}$ byte | $h_1 \in R_{q_1}$ $h_2 \in R_{q_2}$ |
| Private key size | : | $\mathrm{sk} = \frac{2N\log_2 p + N\log_2 q_1}{8}$ byte | $f \in R_p$ $f_p^{-1} \in R_p$ $G \in R_{q_1}$ |
| Packaged key pair size | : | $\mathrm{ct} = \frac{N(\log_2 q_1 + \log_2 q_2 + \log_2 2)}{8}$ byte | $c_1 \in R_{q_1}$ $c_2 \in R_{q_2}$ $K \in R_2$ |

$b = 364$. Similarly, the estimations can be made with $b = 470$ or $b = 496$ for $n = 192$ and $b = 612$ for $n = 256$.

- A dual attack aims to solve the decisional-LWE problem, which provides the obtaining secret key by recovering part of the secret. This attack is made by using the BKZ algorithm in dual lattices. In the concrete hardness assumptions of NIST's PQC standard Kyber (*National Institute of Standards and Technology (NIST), 2023*), dual attacks were not considered since it seems less realistic than the primal attack. Therefore, in the D-NTRU KEM algorithm, the dual attack is not considered since it is much more expensive and impracticable than the primal attack (*Albrecht et al., 2018*; *Liang et al., 2022*).

**Remark 2** *The man-in-the-middle (MITM) attack examination of the D-NTRU KEM is done regarding the distribution parameter of the key generation procedure. In the proposed the D-NTRU KEM, the key polynomials are chosen $f \leftarrow^r L(d_f, d_f - 1)$, $g \leftarrow^r L(d_f, d_f - 1)$ and $r \leftarrow^r L(d, d)$, where $d, d_f, d_g$. Since $d = d_f = d_g$ in the D-NTRU-based schemes (*Wang, Lei & Hu, 2018*), MITM analysis is performed according to the key and message security calculations, presented in Table 2. The main security parameter is obtained by selecting the minimum key and message results.*

**Remark 3** *In the IND-CCA2 security game, when a ciphertext $C = (c_1, c_2)$ and key $K$ is given, it is wanted to determine whether $K$ is generated uniformly random or ciphertext distribution by using decapsulation oracle. A possible attack scenario in checking IND-CCA2 security is as follows:*

**Table 3  Proposed parameter sets.**

| Parameter set | | | |
|---|---|---|---|
| $\kappa = \lfloor ms, ks \rfloor$ | 128 | 192 | 256 |
| Key security (KS) | 128 | 192 | 256 |
| Message security (MS) | 130 | 193 | 257 |
| N | 509 | 677 | 821 |
| $q_1$ | 2,027 | 2,027 | 4,091 |
| $q_2$ | 2,029 | 2,029 | 4,093 |
| $p$ | 3 | 3 | 3 |
| $d$ | 23 | 35 | 48 |
| $b$ | 364 | 470 or 496 | 612 |

**Note:**
 $N$, lattice dimension; $q_1, q_2, p$, moduli values; $d$, sample distribution parameter; $b$, primal attack component.

**Table 4  A comparison for the NTRU/D-NTRU-based IND-CCA2 KEM schemes.**

| Security level | Parameters Schemes | N | q | pk | sk | ct |
|---|---|---|---|---|---|---|
| 128 | ntruhps2048509 | 509 | 2,048 | 699 | 935 | 699 |
| | Ours | | $q_1 = 2,027$ $q_2 = 2,029$ | 1,397 | 1,599 | 1,461 |
| 192 | ntruhps2048677 | 677 | 2,048 | 930 | 1,234 | 930 |
| | Ours | | $q_1 = 2,027$ $q_2 = 2,029$ | 1,859 | 2,127 | 1,943 |
| 256 | ntruhps4096821 | 821 | 4,096 | 1,230 | 1,590 | 1,230 |
| | Ours | | $q_1 = 4,091$ $q_2 = 4,093$ | 2,462 | 2,787 | 2,565 |

*Let the ciphertext of the message M be C and $C' = (c_1' = c_1 + 1, c_2' = c_2 + 1)$. Assume that the first coefficient of $r_2$ was not 1, so that $r_2' = r2 + 1$ is still a ternary noise (this happens with probability at least 1/2). Then $C' = (c_1', c_2')$ is an encryption of M', with noise terms $r_1' = r_1 - f - 1$ and $r_2'$. Although $r_2' = r_2 + 1$, the term $r_1'$ is unpredictable for the attacker as it contains the component belonging to the secret key. Then, even if $c_1$ and $c_1'$ are known, M cannot be obtained from M' since $r_1' \neq r_1 + 1$.*

## COMPARISON

In this section, the proposed parameter set and the comparison analysis of the D-NTRU KEM are presented. The parameter set, given in Table 3, is obtained by adapting the NTRU parameters according to the correctness and security analysis. To compare with the NTRU-based schemes, we developed a python script (*Seyhan, Akleylek & Dursun, 2023*) based on the D-NTRU KEM bounds and the default values of *Chen et al. (2022)*. Table 3 presents the lattice size, modulo, distribution, and the security parameters of proposed the D-NTRU KEM for 128, 192, and 256-bit security levels.

**Table 5 The parameter comparison for the NTRU/D-NTRU-based IND-CCA2 KEM schemes.**

|  | ntruhps (*Chen et al., 2022*) | Ours |
|---|---|---|
| Assumption | NTRU | D-NTRU |
| Lattice dimension (N) | Prime | Prime |
| Modulo value (q) | $2^k$ | $\gcd(q_1, q_2) = 1$ $\gcd(q_1, p) = 1$ $q_1 \rightarrow$ prime |
| Error bound | $\frac{q}{8} - 2 \geq \frac{2N}{3}$ | $q_1 > \phi$ |
| SPD ($L_f$) | T | $L(d, d-1)$ |
| RPD ($L_g$) | $T(q/8 - 2)$ | $L(d, d-1)$ |
| RPD ($L_r$) | T | $L(d, d) \times R_p$ |
| Message distribution | T | $R_p$ |
| IND-CCA2 structure | NAEP padding | One-way encryption function |

Note:
SPD, secret polynomial distribution; RPD, random polynomial distribution; T, ternary polynomials; $T(q)$, the subset of T. $q/2$ coefficient of $T(q)$ is equal to 1, the remaining $q/2$ coefficient is equal to $-1$.

The theoretical basis of the proposed KEM is explained in Table 2. Based on Table 2, the developed script (*Seyhan, Akleylek & Dursun, 2023*) was used to determine the suitable parameters and sizes. By following the message and key security computation, the values $N$ and $d$ are determined for each security level. The twin primes $q_1$ and $q_2$ are chosen to satisfy failure condition $q_1 > 40d - 18$, where $q_1 + 2 = q_2$. The ntruhps (*Chen et al., 2022*) values were selected as a reference for comparison.

By using Tables 2 and 3, the computed components of the D-NTRU KEM are presented in Table 4. In Table 4, the public/secret keys and ciphertext sizes are obtained in bytes using script (*Seyhan, Akleylek & Dursun, 2023*). Since no other D-NTRU-based IND-CCA2 KEM exists in the literature, the comparison can be made with the NTRU-based ones such as ntruhps (*Chen et al., 2022*). According to Table 4, the proposed KEM provides relatively larger key and ciphertext sizes for the same security level. The main parameters such as lattice size, moduli value, error bounds, parameter/message distributions, and security components that cause the differences of compared schemes are expressed in Table 5. Different hard problems and special requirements cause these differences. According to comparison analysis, the proposed method is characterized by the absence of any padding mechanism and arithmetically simple operations.

## CONCLUSION

In this article, we construct a novel D-NTRU-based KEM scheme. It provides a solution to define IND-CCA2 security of the D-NTRU-based encryption, an open problem in *Wang, Lei & Hu (2018)*. The security of the proposed KEM relies on the hardness assumption of the D-NTRU problem. Based on the one-way D-NTRU IND-CPA encryption scheme, IND-CCA2 secure D-NTRU KEM is constructed by following Dent's KEM architecture (*Dent, 2003*). The detailed security analysis is done in the ROM according to modified Dent assumptions for the D-NTRU-based structures. The basic lattice-based attack evaluations are also presented. The proposed KEM is the first IND-CCA2 secure D-

NTRU-based KEM in the literature. It has a simple design and the fact that it does not involve any padding mechanisms. The D-NTRU KEM trivializes the large key and ciphertext sizes. As a future work, we will focus on the D-NTRU-based KEM schemes, including methods such as NAEP padding and their security analysis in the quantum random oracle (QROM) model.

### Funding
This work was supported by TUBITAK under Grant No. 118E312. There was no additional external funding received for this study. The funders had no role in study design, data collection and analysis, decision to publish, or preparation of the manuscript.

### Grant Disclosures
The following grant information was disclosed by the authors:
TUBITAK: 118E312.

### Competing Interests
The authors declare that they have no competing interests. Sedat Akleylek is a PeerJ Section Editor of Cryptography.

### Author Contributions
- Kübra Seyhan conceived and designed the experiments, performed the experiments, analyzed the data, performed the computation work, prepared figures and/or tables, authored or reviewed drafts of the article, and approved the final draft.
- Sedat Akleylek conceived and designed the experiments, performed the experiments, analyzed the data, performed the computation work, prepared figures and/or tables, authored or reviewed drafts of the article, and approved the final draft.

### Data Availability
  No data has been used. This is a theoretical study.

### Supplemental Information
Supplemental information for this article can be found online at http://dx.doi.org/10.7717/peerj-cs.1391#supplemental-information.

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
