# Peer review of "Indistinguishability under adaptive chosen-ciphertext attack secure double-NTRU-based key encapsulation mechanism"

_PeerJ Computer Science, doi:10.7717/peerj-cs.1391_

## Round 0.1 · original submission · Major Revisions

Though one reviewer has recommended minor revision, their concern about lack of security proof entails a major revision. The other reviewer has very clearly stated the need for a major revision, accompanied with a list of concerns - including formal and more rigorous proofs of security, a bar that needs to be satisfied for a paper on cryptography to be accepted.

The authors need to address all these issues when they revise the manuscript.

Reviewer 1 ·

Basic reporting

This submission presents an IND-CCA secure KEM algorithm based on the D-NTRU lattice public key cryptosystem. I think the main contribution of the paper is a natural padding for IND-CPA secure D-NTRU cryptosystem. The technical is technically sound. However, my main concern is about the security evaluation of the IND-CCA KEM algorithm.

In fact, there exist some security evaluation models for lattice public key cryptography, for example, the primal attack and the dual attack models. Under the suggested parameters in Table 3, I think that the security was overestimated. The reason for the authors to obtain these security evaluations may be because the authors just borrow the security results from the original paper of D-NTRU. The original paper just consider the message security and the key security based on brute-force attacks. So I suggest the authors to re-evluate the securities. And also, the comparisons should be re-performed in order that the comparison results were made under the same security levels.

There are also some minor issues that need to be addressed.
1. Some typos. For example, in table 1, The degree of T is at most N-2, where I think that it should be N-1.
2. Why the security proof was completed from the one-way security of D-NTRU, but not the stronger IND-CPA security?

Experimental design

I have no comments in this area.

Validity of the findings

I have no comments in this area.

Additional comments

I have no comments in this area.

Reviewer 2 ·

Basic reporting

The English text is good and comprehensible enough in most places, but would benefit from editing by a fluent English-speaker. (Examples: page 1 line 56 extra “the”; page 1 line 45 moduli not modulos; page 2 line 66 plural incorrect, page 3 line 88 the commas must be removed or else it says all KEM schemes depend on IF or DLP; etc).

The text would also benefit from thorough proofreading. Examples: page 1 line 44 based on reduction modulo *two relatively prime positive integers* (text in * * missing); page 2 line 48 sentence incomplete, In Table 1, T_+ does not make sense (correlation with what?), etc.
The language of the article is courteous and professional.
The general overview provides some context of the background of the topic and otherwise does include many key moments in the history of NTRU. However, it has been known since a Diploma project in by Roman Kouzmenko in 2006 that CTRU is completely vulnerable to a simple linear attack, but this is not known to the author here. The abstract refers to padding and the NAEP transformation but this is not mentioned in the body of the text, even though it is a much more relevant comparison and discussion.

The article structure is professional. The figures and tables are a problem. Figure 5 is decoration and does not contribute to the understanding in the text. Many figures seem to have been taking from other sources (they are typset differently). Figures 2 and 3 seem to be directly copied from another paper, which is not acceptable; not all notation is defined and the algorithms are not explained in the text. The font size is too small; the author should recreate the algorithms without the strange layout (Alice, Bob and a middle column) that are not needed and do not correspond well with what is meant. Figure 6 is similarly illegible and again should have been part of the text. Figure 3 mixes R_q and mod q notation inappropriately.

To explain the algorithms, the author should define all variables and terms. Example: h[i] in Figure 2 is undefined; q is not defined. Once the algorithm has been summarized, with explanations as to the significance of the choices and steps, one can include a table for convenient reference.

The article is not self-contained. Several theorems are stated without proof or reference and the key steps of the algorithm are not discussed in any detail. Table 1 is not readable without intimate knowledge of NTRU. Some statements in Table 1 are imprecise: the infinity norm cannot be defined on an element of Z_q because finite rings have no well-defined absolute value; instead one can define it on an element of Z (as in NTRU, one identifies the reduction mod q with an integer in order to switch moduli). Concepts such as decryption failure are undefined.


In Section 2, the author give Lemma 1; but Lemma 1 is a very complicated way of saying that the polynomial has at least one coefficient that is not 0 or 1. No proof or reference. Theorems 1, 2 and 3 are similarly stated separately but with no proof, explanation or reference. Citation should be given directly to its origin, and it would be better to combine them all into one well-worded theorem. The intervening text contributes no understanding to the result. Only Theorem 1 is used elsewhere in the paper; the others do not contribute to the explanation of the result. In Section 2.2, the definitions are too imprecise to be useful and the theorems again are stated without proof and without appropriate reference. Again, these results are not used in the rest of the paper; the space would be better spent presenting D-NTRU.
The paper is indeed a unit of publication, presenting a complete story.
The main results are presented in Section 3 and 4. The hash functions used in the Scheme are not defined; that “kdf allow a random number to be formatted as a key by removing the algebraic relation between inputs” makes no sense.

The proof of correctness in Section 3.1 needs work. Do not start a proof with the equality that you want to prove; remove r_2= from the beginning of the sequence and instead start with the expression on the right, then do the derivations, and conclude that the expression simplifies to r_2. (Etc)

The colours in the derivation are very nice; they significantly add to the understanding of this complicated derivation. The statement of where the hypothesis Theorem 1 is used is correct and helpful, but truly, nowhere in this paper does the author explain the key secret of NTRU, that is, the delicacy of evaluating moduli with respect to two different values. It is also unclear why D-NTRU is better and why it needs twin primes.

The explanation of the security game is standard and nicely done. One line 217, it is incorrect to say “r2: Otherwise, A cannot gain any advantage in this game. That is false; A has not gained an advantage in this round. The rest of the sentence should not be part of r2; the rest of the sentence is about what you can say after A has played many rounds of the game and you have deduced the probability that she wins.

The main proofs are in 3.2, but are very unsatisfactory. For example, Assumption 1 (line 236) is a very simple statement about probability, but the author states it as an assumption. Grammatically, stating it as an assumption means that you don’t know if the hypothesis implies the conclusion. But of course you can prove that the hypothesis implies the conclusion. So if you wanted to just remind us of this cute fact in probability, you should state it as a lemma and quickly prove it. If on the other hand you thought you were stating the hypothesis as an assumption, then the statement is in the wrong place as you have not yet defined Y_1 or Y_2.

The work of the proof is missing. At line 247, the author simply says that the difference in probabibility is negligible. But this is the whole point --- you want to argue that even an attacker who has clever ideas can’t do better with one ciphertext than the other, but instead you are assuming the result you want to prove.

Because the hash functions are modeled as random oracles, the rest of the proof is that because the attacker can’t guess the ciphertext by guessing randomly, he has no advantage on the protocol. The hash functions are magical and it would be interesting to use real ones with some actual bounds on collisions.

Several statements in the proof are incorrect; eg the total number of queries is not the fraction 1/q + 1/(k+m). Remark 2 is presumably where the actual hard part of the proof is discussed, but only in one trivial example; it is this kind of attack that a serious security game would need to argue against.

Figure 6 is not referenced in the text except for a minor point and does not aid in understanding.

The parameter sets in Table 2 seem to say that the security of NTRU is based entirely on the size of its key space. But lattice attacks are much more efficient than brute force attacks? Where is this from?

Table 4 does not say what level of security was chosen for the other 2 KEMs, so it is not fair to compare. Table 5 does not contribute to understanding; the terms are not defined and it is unclear what the table is trying to show.

Experimental design

This is original primary research and the question is meaningful and relevant, and fills an identified knowledge gap (how to produce a KEM from D-NTRU with IND-CCA2 security).

The investigation is not at the level of rigour required in cryptography; it is however an entirely valid approach to creating such a KEM. The author did not explore enough of the modification they introduced to convince the reader that their modification introduced no weaknesses and yet also created a randomly distributed key even though the original D-NTRU could not.

Validity of the findings

This work meets the standards of introducing a KEM, though it does not offer appropriate comparisons of performance, efficiency and does not provide convincing argument of validity.

The conclusions should be appropriately stated and correspond to the claims made.

---

## Round 0.2 · Major Revisions

The paper is very poorly written, and bogged down by the poor writing, one of the reviewers has not even reviewed the technical content. I checked personally parts of the text, and the quality of English, as well as notational consistency, etc. are all of serious concern. As such, even though one of the reviewer has recommended acceptance, taking into account the other reviewer's detailed comments, I am recommending a major revision.

Please use a proof checker with good proficiency in English, and also fix all other kind of inconsistencies, e.g., notational and factual (e.g., NTRU being in the NIST shortlist is an outdated information).

If the next revision is not deemed adequate for acceptance as is, I may have to recommend a rejection without further iterations of revisions. As such, please carry out the revision very carefully.

Reviewer 1 ·

Basic reporting

All my concerns were fully addressed, so I commend to accept the revised submission for publication.

Experimental design

I have no further comments.

Validity of the findings

The construction of IND-CCA2 D-NTRU KEM is interesting.

Additional comments

I have no further comments.

Reviewer 2 ·

Basic reporting

line 16-17: you propose something for DNTRU, not for something "such as DNTRU" ? This sentence should be removed; it is repetition; the last sentence says it again and provides some content.

line 24: incorrect citation form; the paper is by W Diffie and M Hellman, not Hellman et al.

line 28-29: "in polynomial time in quantum computing power" -> "in polynomial time on a sufficiently large quantum computer"

lines 35-39: you fail to say that none of the NTRU family was chosen as a finalist nor for standardization in the 4th round of the NIST exercise

line 44: independence not independency

lines 44-49: this is generic and has no content. What is the relationship of p and q to make them "independent"; what do you mean "elementary probability theory is used in the decryption phase of NTRU"? (False) And NO if there are many short vectors close in length to the shortest vector then these other ones can ALSO be used as a key; that is NOT the correct interpretation of why this should be considered hard.

Figure 1: should be removed. This is neither a complete literature review nor does it add to the discussion.

lines 52-59: this should be deleted, and also CTRU deleted from the table (where you incorrectly say it was based on F_2 rather than F_2[T]). It was proven to be totally insecure, so the ONLY value in mentioning it would be to explain WHY it was a bad idea to replace Z with F_2[T]. It is complete nonsense to say "until 2006 it was secure".

line 65: Either use more standard citation style, or incorporate the citation in your text; you repeat the author's names. Moreover, these authors completely changed the ring; the change of distribution is a minor comment in the abstract of this cited paper but hardly the most significant piece of it.

lines 80-82: Again, this current paper was submitted well AFTER it was known that the NTRU proposal had not been selected; it is inappropriate to suggest otherwise.

lines 99-100: Same problem. This current paper was submitted in november 2022 and the authors revised it in early 2023; this oversight is unacceptable.

line 107: "given in the random oracle model" (not "following ROM assumptions")

Table 1: x_p^{-1} is inverse of x mod p, not mod X

Line 137: C-NTRU and D-NTRU were produced in the SAME PAPER, and in fact the authors Wang et al only created C-NTRU to help explain how D-NTRU works. This is not communicated properly here.

Line 141-144: reduce the definition to the actual definition, and save commentary for afterwards.

Experimental design

Line 145: what is "conclusion 1"?

Line 147: Better statement: IF these conditions hold THEN decryption will not fail.

Line 155: But this condition will virtually never hold: G is given by p^{-1}\otimes g_{q_1}^{-1} and g is chosen to have small coefficients. In fact, the original authors Wang et al make it clear that one must ensure its coefficients are larger than this or else each encryption will be easy to crack.

This error at line 155 is not acceptable. In fact, the original authors Wang et al get it right; the current authors do not understand D-NTRU.

In the proof of theorem one, the last line is incorrect. You mean to say that the previous line (<q_1/2) holds BECAUSE q_1>delta.

Line 164-165: this is false, or at least not justified by referring to Theorem 1. In the original article by Wang et al, the authors prove a different statement, one that uses Theorem 1 to infer that a lack of decryption failure implies that the message is uniquely determined. This is NOT what is going on here.

Line 216-218: This is stated in the wrong direction. If Problem 1 reduces to Problem 2, then if Problem 2 is hard, so is Problem 1. In fact, Wang et al prove the polynomial equivalence of each of these pairs of problems.

When citing specific results from another paper, use \cite[Theorem 3]{Wangetal} to help the reader find the correct and full statement.

Line 252: equation (3) --- REFER to your Theorem 4! Without it, the derivation is false; you should put Theorem 4 before the argument of correctness as it is ESSENTIAL.

equation (4) --- at the end, for calculating M, the mod q_2 terms are missing.


I stopped reading here.


The reference list is unacceptably sloppy. Many references are incorrectly formatted, with capitalizations missing. Line 438 incorrectly attributes the seminar Diffie-Hellman paper as "Hellman et al". Many references are incomplete.

Validity of the findings

I did not make it to the rest of the paper.

Additional comments

I appreciate the effort the authors made to address the comments in the first review; this paper is much better written and more complete. It still contains, however, too many mistakes to be considered for publication.

---

## Round 0.3 · accepted · Accept

The write-up of the revised document is reasonably improved, though there are a few remaining typos. There are no outstanding technical deficiencies remaining.